# Uroguanylin Improves Leptin Responsiveness in Diet-Induced Obese Mice

**DOI:** 10.3390/nu11040752

**Published:** 2019-03-30

**Authors:** Cintia Folgueira, Daniel Beiroa, María Jesús González-Rellán, Begoña Porteiro, Edward Milbank, Cecilia Castelao, María García-Palacios, Felipe F Casanueva, Miguel López, Carlos Diéguez, Luisa M. Seoane, Rubén Nogueiras

**Affiliations:** 1Department of Physiology, CIMUS, Universidade de Santiago de Compostela - Instituto de Investigación Sanitaria, 15782 Santiago de Compostela, Spain; cintiafolgueira@gmail.com (C.F.); danielbeiroa@gmail.com (D.B.); chusa.gzlz.rellan@gmail.com (M.J.G.-R.); begoporteirocouto@gmail.com (B.P.); edmilbank@gmail.com (E.M.); m.lopez@usc.es (M.L.); carlos.dieguez@usc.es (C.D.); 2Grupo Fisiopatología Endocrina, Instituto Investigación Sanitaria Santiago de Compostela (IDIS), Complejo Hospitalario Universitario Santiago de Compostela (CHUS/SERGAS), 15706 Santiago de Compostela, Spain; ceciliacastelao@hotmail.com (C.C.); maria.garcia.palacios@gmail.com (M.G.-P.); 3CIBER Fisiopatología Obesidad y Nutrición (CIBEROBN), Instituto Salud Carlos III, 15706 Santiago de Compostela, Spain; felipe.casanueva@usc.es; 4Department of Pediatric Surgery, Universidade de Santiago de Compostela, 15706 Santiago de Compostela, Spain; 5Laboratorio Endocrinología Molecular y Celular, Universidade de Santiago de Compostela, 15706 Santiago de Compostela, Spain

**Keywords:** uroguanylin, leptin, food intake, body weight, hypothalamus

## Abstract

The gastrointestinal-brain axis is a key mediator of the body weight and energy homeostasis regulation. Uroguanylin (UGN) has been recently proposed to be a part of this gut-brain axis regulating food intake, body weight and energy expenditure. Expression of UGN is regulated by the nutritional status and dependent on leptin levels. However, the exact molecular mechanisms underlying this UGN-leptin metabolic regulation at a hypothalamic level still remains unclear. Using leptin resistant diet-induced obese (DIO) mice, we aimed to determine whether UGN could improve hypothalamic leptin sensitivity. The present work demonstrates that the central co-administration of UGN and leptin potentiates leptin’s ability to decrease the food intake and body weight in DIO mice, and that UGN activates the hypothalamic signal transducer and activator of transcription 3 (STAT3) and phosphatidylinositide 3-kinases (PI3K) pathways. At a functional level, the blockade of PI3K, but not STAT3, blunted UGN-mediated leptin responsiveness in DIO mice. Overall, these findings indicate that UGN improves leptin sensitivity in DIO mice.

## 1. Introduction

Obesity is characterized by a deregulation of numerous pathways involved in the energy homeostasis modulation. In order to maintain the body weight at stable levels, the central nervous system (CNS) is constantly informed about the status of the organism’s energy stores by peripheral systems through complex pathways. Among others, the gastrointestinal-brain axis has been described as a key player in all this metabolic homeostasis regulation [1]. Indeed, the large spectrum of peptides secreted by the gastrointestinal tract regulates the two sides of the energy balance equation (i.e., food intake and energy expenditure).

Uroguanylin (UGN) has been recently proposed to be one of the members of this gut-brain axis implicated in body weight regulation [2,3,4,5,6]. UGN is a 16 amino-acid peptide secreted by the duodenal epithelial cells under its precursor form – prouroguanylin (pro-UGN) –, the pro-UGN is cleaved to give rise to the active form of the peptide which will bind the transmembrane receptor guanylin cyclase 2C (GUCY2C) [7]. Interestingly, it has been shown that UGN could bind GUCY2C at a hypothalamic level inducing the activation of anorexigenic pathways [6]. Although, subsequent studies failed to confirm UGN effects on food intake [8,9], we have recently reported that UGN centrally administered induced a body weight decrease on diet-induced obese (DIO) mice by increasing the energy expenditure in a food intake-independent manner [10].

Intestinal and plasmatic UGN levels are dependent on the nutritional status [11]: expression of UGN could be (i) increased concomitantly with food intake [6], (ii) reduced under fasting conditions and (iii) recovered following refeeding [11]. Interestingly, it has been demonstrated that this UGN nutritional status-based modulation was dependent on leptin. Leptin is an adipocyte derived hormone mainly implicated in the enhancement of metabolism and food intake reduction [12,13]. Leptin circulating levels are positively correlated with the adipose tissue mass [12]. Although obese patients exhibit high levels of leptin, they also appear to be resistant to its central hypothalamic effects [14,15,16,17]. The exact mechanisms still remain unclear, however this leptin resistance seems to be mainly due to a decrease in leptin transport across the blood-brain barrier and to an impairment of several leptin receptor-signaling pathways [17,18,19,20]. However, some studies have demonstrated that leptin sensitivity could be improved using pharmacological agents [21,22,23], suggesting that leptin administration could be a valuable target when used in combination with other hormones or peptides [24]. Thus, taking into account that the regulation of UGN levels by the nutritional status is dependent on leptin, and that centrally administrated UGN reduces body weight in DIO mice, within this study we wanted to evaluate whether UGN could restore leptin sensitivity in DIO mice models.

## 2. Materials and Methods

### 2.1. Animals and Diets

Swiss male mice (20–25 g, 8–10 weeks old) were housed in individual cages under controlled conditions of illumination (12:12-h light/dark cycle), temperature and humidity. Mice were allowed *ad libitum* access to water and a high fat diet (HFD) (Research Diets 12451; 45% of calories from fat, 4.73 Kcal/g, Research Diets, New Brunswick, NJ) for 13 weeks. 7–11 animals per group were used.

The food intake and body weight were measured daily. Animals were sacrificed by decapitation; the brains were rapidly removed and immediately frozen on dry ice and kept at −80 °C until their analysis. All experiments and procedures involved in this study were reviewed and approved by the Ethics Committee of the USC, in accordance with the European Union normative for the use of experimental animals.

### 2.2. Treatments and Surgeries

Mice were anesthetized by an intraperitoneal injection of ketamine (8 mg/kg) and xylazine (3 mg/kg). Intracerebroventricular (ICV) cannulae were implanted stereotaxically in mice, as previously described [25,26,27]. Animals were ICV injected with vehicle (saline), UGN (25 μg/mouse, Bachem, Bubendorf, Switzerland) and/or leptin (3 µg/mouse, kindly provided by Dr A. F. Parlow, National Hormone and Peptide Program, Harbor- UCLA Medical Center, Torrance, CA, USA). To perform the molecular analysis, the mice were sacrificed 15 min after the leptin ICV administration, and the brains were immediately removed, frozen and stored at −80 °C.

To study the STAT3 pathway, the mice were fasted overnight and then ICV injected with either vehicle (saline) or a peptide inhibitor of STAT3 (STAT3 PI) (75 pmol/mouse, Calbiochem, San Diego, CA, USA) [28]. Thirty minutes later, the mice received either ICV vehicle (10 mM NaHCO_3_, pH 7.9) or leptin (3 µg/mouse) as previously described [28]. For the analysis of phosphatidylinositide 3-kinase (PI3K) pathway, the mice were fasted overnight and then received an ICV infusion of either vehicle (DMSO) or an inhibitor of PI3K (LY294002) (1 nmol/mouse, Sigma-Aldrich, St. Louis, MO, USA). Twenty minutes later, the mice received either an ICV vehicle (10 mM NaHCO_3_, pH 7.9) or leptin (3 µg/mouse) [28].

### 2.3. Dissection of Brain Areas

After the sacrifice, the brains were removed and immediately frozen and stored at −80 °C until further processing. Then, the brain was placed in a brain matrix with the ventral surface under a dissecting microscope. The mediobasal hypothalamus was removed from the whole hypothalamus by cutting between the rostral and caudal limits of the median eminence parallel to the base of the hypothalamus and 0.5 mm to each lateral side of the median eminence [27,29,30].

### 2.4. Real Time Polymerase Chain Reaction (PCR)

The specificity of the mediobasal hypothalamus isolation was evaluated by measuring the mRNA levels of *Agrp* and *Pomc*. RNA was isolated using the TRIzol Reagent (Invitrogen, Carlosbad, CA, USA) according to the manufacturer´s instructions [10]. The extracted total RNA was purified with the DNase treatment using a DNAfree kit as a template (Ambion; Thermo Fisher Scientific, Grand Island, NY) to generate the first-strand cDNAs using a High-Capacity cDNA Reverse Transcription Kit (Applied Biosystems, Foster City, CA, USA). Quantitative real-time PCR was performed using a StepOnePlus Instrument with specific TaqMan quantitative RT-PCR primers and probes. The oligonucleotide-specific primers are listed in Table 1. Hypoxanthine phosphoribosyltransferase (*Hprt*) was used as an endogenous control, and the expression levels in the sample group were relatively expressed to the average of the control group.

### 2.5. Western Blotting

The mediobasal hypothalamus was isolated with a brain dissection block, as previously described [27,29,30,31]. Tissues were homogenized using a TissueLyser II (Qiagen, Tokyo, Japan) in fresh RIPA buffer (containing 200 mMTris/HCl (pH 7.4), 130 mM NaCl, 10%(*v*/*v*) glycerol, 0.1%(*v*/*v*) sodium–dodecyl sulfate (SDS), 1%(*v*/*v*) Triton X-100, 10 mM MgCl2) containing anti-proteases and anti-phosphatases (Sigma-Aldrich, St. Louis, MO, USA). The tissue lysates were centrifuged for 30 min at 18,000 *g* in a microfuge at 4 °C. The mediobasal hypothalamus total protein lysates were subjected to SDS-polyacrylamide gels (SDS–PAGE), then electrotransferred on a PVDF membrane and probed successively with the following antibodies: Phospho-PI3K P85, PI3K P85, phospho-STAT3 and STAT3 (Cell Signaling, Danvers, MA, USA); β-actin (Sigma-Aldrich, St. Louis, MO, USA) after blocking the membranes with 5% BSA blocking buffer. For the protein detection, we used horseradish-peroxidase-conjugated secondary antibodies (Dako Denmark, Glostrup, Denmark). Specific antigen-antibody bindings were visualized using chemiluminescence method according to the manufacturer´s instructions (Pierce ECL Western Blotting Sustrate, Thermo Fisher Scientific, Grand Island, NY, USA). Values were expressed in relation to β-actin.

### 2.6. Blood Biochemistry

Blood samples were harvested from the saphenous vein and the levels of leptin were determined using a Leptin Mouse enzyme-linked immunosorbent assay (ELISA) (Millipore, Catalog number EZML-82K).

### 2.7. Statistical Analysis

Results are expressed as mean ± standard error of the mean (SEM). The GraphPad Prism Software Version 5.0 (GraphPad, San Diego, CA, USA) was used for the data analysis. The number of animals used is listed in the figure legends. Statistical analysis was performed using a one-way analysis of variance followed by a post hoc multiple comparison test (Bonferroni test) for multiple comparison tests. For two population comparisons, an unpaired *t* test (two-tailed) was used. A *p* value less than 0.05 was considered statistically significant.

## 3. Results

### 3.1. Central UGN Administration Restores Leptin Sensitivity in Diet-Induced Obese (DIO) Mice

In order to evaluate whether UGN could improve leptin sensitivity, the DIO mice were first ICV injected with vehicle or UGN (25 µg/day during four days, a dose known to induce body weight decrease in DIO mice) (Figure 1A). In accordance with the previous data we have obtained in 2016 [10], the central administration of UGN in DIO mice induced a body weight decrease (Figure 1B) without modifying food intake (Figure 1C) and this during three days. At the end of the first three days, the two groups of mice were fasted overnight before being injected either with vehicle or leptin at a dose of 3 µg, known to induce a decrease of feeding and bodyweight in chow diet fed rodents [28] (Figure 1A). As shown on Figure 1D,E, the ICV administration of leptin in non-UGN pretreated mice did not induce any changes in body weight (Figure 1D) or food intake (Figure 1E,F). However, in an interesting way, when administrated in obese mice previously treated with UGN for four days, leptin induced a significant decrease in body weight (Figure 1D) as in food intake after 6 h (Figure 1E) and 24 h (Figure 1F). Leptin was also ICV injected in chow diet fed lean mice, and we were able to observe that leptin at the dose of 3 µg induced a decrease in body weight (Figure 1G) and in food intake after six and 24 h (Figure 1H–I). Taking into account all these results, we were able to demonstrate that a central administration of UGN could restore leptin responsiveness in DIO mice.

### 3.2. Central UGN Administration Restores Hypothalamic Leptin-Induced Signaling Pathways in DIO Mice

In DIO mice, the leptin receptor-induced intracellular signaling pathways have been shown to be impaired, including phosphatidylinositol 3-kinase (PI3K) and signal transducer and activator of transcription 3 (STAT3) pathways [12,13]. Therefore, after harvesting the hypothalamus of the different mice and groups, we have analyzed whether these hypothalamic signaling pathways could be affected by UGN. Interestingly, we were able to observe that the separated single UGN or leptin treatments in DIO mice models did not affect the levels of the total or phosphorylated form of PI3K, neither the ones of total and phosphorylated STAT3 (Figure 2A). However, the DIO mice pretreated with UGN during four days then injected with leptin exhibited increased mediobasal hypothalamic levels of the total and phosphorylated forms of PI3K, as well as a tendency to increased levels of phosphorylated STAT3 (Figure 2A). To evaluate the specificity of the isolation of the mediobasal hypothalamus (MBH), we have measured the expression of *Agrp* and *Pomc*—specific mediobasal hypothalamic markers—and interestingly, *Agrp* and *Pomc* mRNAs were only detected in the MBH and not in the lateral hypothalamus (LHA) (Figure 2B). All together, these data suggest that UGN restores hypothalamic leptin-induced signaling pathways in DIO mice.

### 3.3. Blockade of STAT3 does not Affect UGN-Mediated Leptin Responsiveness

Taking into account that both phosphorylated STAT3 and phosphorylated PI3K pathways were activated in the hypothalamus after the co-administration of UGN and leptin, we next performed functional studies to address if these intracellular pathways could mediate the restoration of leptin sensibility in DIO mice. Surprisingly, the ICV single treatment with leptin or combined with the inhibitor of STAT3 (STAT3 PI) did not modify either the food intake or the body weight of DIO mice (Figure 3A,B). The same experimental protocol was realized on the UGN-pretreated mice and as expected, the food intake (Figure 3A) and the body weight (Figure 3B) were decreased. However, the inhibition of STAT3 pathway did not modify the metabolic response of these mice, exhibiting the same decrease in food intake and body weight (Figure 3A,B) as the other groups. Therefore, these data indicate that STAT3 is not playing an essential role in the mediation of UGN effect on leptin sensitivity.

### 3.4. Blockade of PI3K Blunts UGN-Mediated Leptin Responsiveness

We next blocked the central PI3K pathway in DIO mice by administering a PI3K inhibitor, LY294002. As observed in the previous experiment, the ICV treatment with leptin or LY294002 did not modify the food intake or body weight of DIO mice (Figure 4A,B). However, the DIO mice pretreated with UGN and injected with leptin exhibited a decrease in food intake (Figure 4A) and body weight (Figure 4B), reflecting the restoration of leptin sensitivity. However, when the mice were injected with a combined treatment composed of UGN, leptin and LY294002, the food intake (Figure 4A) and the body weight (Figure 4B) were restored to similar levels as the ones observed in the DIO mice injected with the vehicle. Therefore, the present data demonstrate that UGN centrally administered fails to restore leptin sensitivity in DIO mice when PI3K is blocked.

### 3.5. Central UGN Restores Leptin Sensitivity Independently of Body Weight Changes and Leptin Levels in DIO Mice

Later on, we wanted to evaluate whether the effects of UGN on leptin sensitivity were due to a direct action of UGN, rather than being a consequence of the reduction of leptin levels induced by the body weight decrease. Therefore, to obtain reliable results on UGN effects, the same experiments were repeated using weight-matched DIO mice (i.e., the initial body weights were defined according to the body weight measured after three days of UGN treatment), giving rise to five different groups: (i) control group injected by vehicle (DIO+Vehicle), (ii) mice ICV perfused by UGN (25 µg/day during for four days) (DIO+UGN), (iii) control weight-matched mice receiving vehicle (DIO-Weight- matched +Vehicle), (iv) weight-matched mice receiving leptin at a dose of 3 µg (DIO-Weight-matched+Leptin) and (v) weight-matched UGN-pretreated mice receiving leptin at a dose of 3 µg (DIO-Weight-matched +Leptin).

As expected, the central ICV administration of UGN for three days induced a body weight decrease compared to the vehicle perfused control group. Interestingly, the injection of leptin in the weight-matched control mice did not induce any modifications of the body weight. However, the injection of leptin in mice that had been pretreated with UGN for three days exhibited a significant body weight decrease (Figure 5A). As demonstrated before, the body weight decrease (Figure 5B) induced by UGN was not associated to anorexia (Figure 5C). Interestingly, the chronic UGN injection of weight-matched mice during three days induced the same metabolic phenotype response: A body weight decrease (Figure 5D) without any changes in food intake (Figure 5E) as observed in their UGN pretreated control mice. As noticed before, the ICV administration of leptin in weight-matched mice did not modify the body weight (Figure 5F) and the food intake (Figure 5G,H) of the treated mice. However, as expected, injected into UGN-pretreated weight-matched mice, leptin at a dose of 3 µg induced a significant decrease in body weight (Figure 5F) and food intake at 6 and 24 h (Figure 5G,H). To complete this analysis, we have also evaluated the circulating levels of leptin, and we were able to observe that the ICV administration of UGN induced a significant decrease of leptinemia (Figure 5I), consistent with the associated fat mass reduction. Moreover, the circulating levels of leptin were found to be reduced in weight-matched DIO mice compared to their control (Figure 5I), this observation being in accordance with their lower body weight. Overall, these findings indicate that the weight-matched DIO mice also exhibit a leptin resistance, and that UGN is able to induce a body weight decrease in the different groups of mice, suggesting that UGN effects on leptin sensitivity are independent on the body weight and on the leptin levels.

## 4. Discussion

In spite of the widely described effects of leptin on decreasing food intake and body weight on different animal models, the use of leptin as a therapeutic agent in an obesity driven context remains inefficient [14]. Due to the elevated circulating levels of leptin detected in DIO mice and obese human patients inducing a desensitization of the leptin receptor (LepR), the hypothalamic intracellular pathways could be impaired and deregulated [16,17], explaining the lack of therapeutic efficiency of leptin to decrease the body weight. The present study provides mechanistic evidences that the leptin sensitivity can be improved by UGN, inducing a decrease in food intake and body weight in DIO mice. We were able to demonstrate that the UGN effects on leptin sensitivity were mediated by the PI3K pathway while being STAT3-independent, confirming that the leptin resistance can be due to an intracellular signaling defect in leptin-responsive hypothalamic neurons that lies upstream of STAT3 activation [14].

The key role of UGN in the regulation of energy homeostasis through the gut-brain axis has lately acquired great relevance [4,5,32]. In humans, UGN stimulates lipolysis in adipocytes [32] and the UGN mRNA expression is increased in obese patients following bariatric surgery [9]. Moreover, the circulating levels of pro-UGN are decreased in obese adults [33] and obese female adolescents [34] compared to lean subjects. In rodent models, the chronic central administration of UGN reduced the body weight in DIO mice through an increase of energy expenditure [10]. UGN levels are regulated by the nutritional status and this phenomenon is dependent on leptin [11]. Moreover, *ob/ob* mice, a model of genetic obesity characterized by a lack of leptin, show very low levels of UGN, recovering normal levels following a leptin treatment [11]. Given the strong interaction between UGN and leptin, and that both hormones require hypothalamic mechanisms to exert their catabolic actions, it was plausible to hypothesize that these two systems were likely interacting at a hypothalamic level. Our results showing that DIO mice centrally infused with UGN exhibit enhanced leptin responsiveness reducing both feeding and body weight, support this hypothesis.

It is widely known that the activation of the leptin receptor in the hypothalamus induce the modulation of the JAK2/STAT3 and PI3K/AKT pathways [17,35,36] and they have been extensively studied as key molecular determinants of hypothalamic leptin resistance [17,37,38]. As these signaling pathways are blunted in obesity, further research efforts have been made to discover therapeutic agents to improve leptin sensitivity focusing on JAK-STAT and PI3K-Akt-FoxO1. Through the years, some efficient agents have been discovered, such as amylin, which restores leptin responsiveness in both obese rodents and patients [21], as well as FGF21 [22] and GLP-1/glucagon co-agonism [23]. The PI3K/AKT pathway is also activated by UGN since the guanylate cyclase activators modulate AKT signaling in intestinal cells, cardiomyocytes and neurons [39,40,41]. This had led us to hypothesize that UGN could play a key role in restoring leptin sensitivity in DIO mice. Accordingly, the present findings demonstrate that the central administration of UGN enhances leptin-induced phosphorylation of PI3K and STAT3 in the mediobasal hypothalamus of DIO mice. More specifically, UGN requires PI3K but not STAT3, to improve leptin sensitivity. Lastly, the improvement of leptin sensitivity in DIO mice has been shown to be independent of UGN-induced body weight loss or UGN-reduced leptin levels, since the weight-matched DIO mice did not display any metabolic response to single treatment of leptin. However, this study remains mostly pharmacological, and it is likely that the dose of central administrated UGN used here is probably higher than the amount of endogenous UGN reaching the brain under physiological conditions. Further studies using genetically modified mice lacking UGN or GUCY2C peripherally injected with UGN are needed to elucidate whether physiological levels of UGN would be sufficient to improve leptin sensitivity.

In summary, the present results indicate that UGN is a new factor in the improvement of leptin sensitivity in DIO mice and its effects are dependent on the activation of PI3K pathway in the hypothalamus. These results support that pharmaceutical targeting of UGN in combination with leptin administration could be a new strategy to reverse obesity. 

## Figures and Tables

**Figure 1 nutrients-11-00752-f001:**
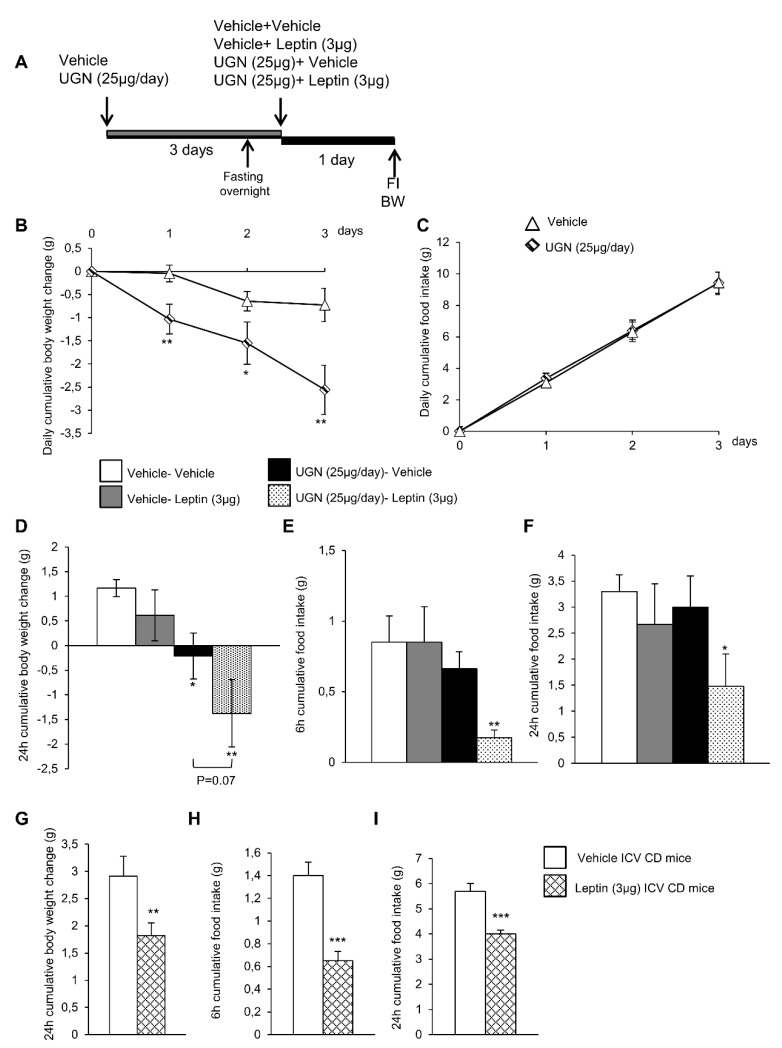
Schematic representation of the in vivo protocol (**A**). Effect of chronic Intracerebroventricular (ICV), uroguanylin (UGN) (25 µg/mouse) injection for four days in DIO mice on body weight (BW) (**B**) and food intake (FI) (**C**). Effect at 24 h of ICV leptin injection (3 µg/mouse) (the mice being pretreated with UGN (25 µg/mouse/four days) and submitted to fasting the night before leptin injection) body weight (**D**), and food intake at 6 h (**E**) and 24 h (**F**). Effect at 24 h of ICV leptin injection (3 µg/mouse) in lean mice fed a chow diet (**C**,**D**) on body weight (**G**) and food intake after 6 h (**H**) and 24 h (**I**). Values are represented as mean ± standard error of the mean (SEM); *n* = 8–17 animals per group. * *p* < 0.05; ** *p* < 0.01; *** *p* < 0.001 vs. vehicle.

**Figure 2 nutrients-11-00752-f002:**
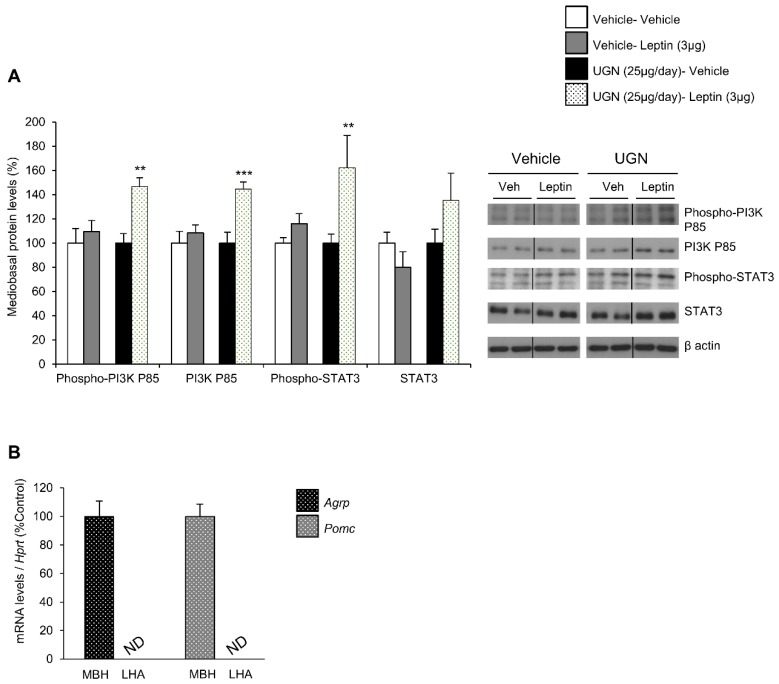
Evaluation of mediobasal hypothalamic levels of Phospho-PI3K P85, PI3K P85, Phospho-STAT3, STAT3 following an ICV leptin injection (3 µg/mouse) (the mice being pretreated with UGN (25 µg/mouse/four days) and submitted to fasting the night before leptin injection). The mice were sacrificed and the tissues collected 15 min after leptin injection. Values are expressed in relation to β-actin levels. Dividing lines indicate splicing within the same gel (**A**). mRNA expression of *Agrp* and *Pomc* in mediobasal and lateral hypothalamus (**B**). Values were expressed relatively to Hypoxanthine-guanine phosphoribosyltransferase (*Hprt*) levels. ND: non detected. Values are mean ± SEM of 8 animals per group. ** *p* < 0.01, *** *p* < 0.001 vs vehicle.

**Figure 3 nutrients-11-00752-f003:**
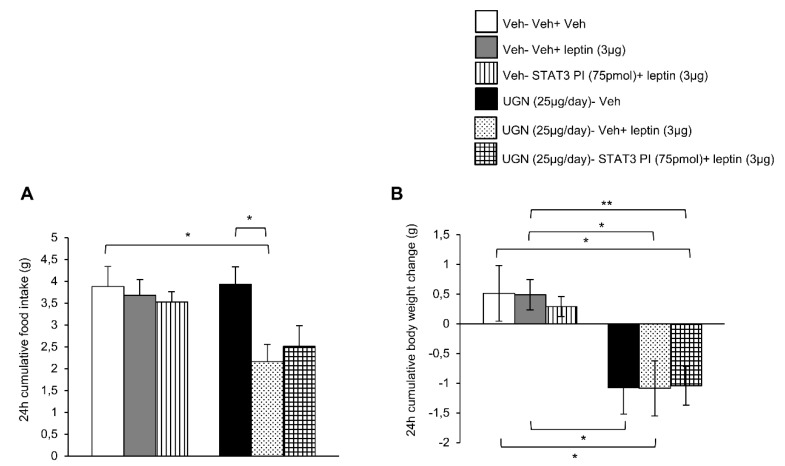
Evaluation of the blockade of STAT3 pathway in the response in 24 h cumulative food intake (**A**) and body weight change (**B**). The mice were perfused with UGN (25 µg/mouse) for four days, submitted to fasting overnight and then injected with leptin (3 µg/mouse), an inhibitor of STAT3 (STAT3 PI) (75 pmol/mouse), or by the combination leptin/STAT3 PI. Values are represented as mean ± SEM; *n* = 10–11 animals per group. * *p* < 0.05, ** *p* < 0.01 vs. vehicle.

**Figure 4 nutrients-11-00752-f004:**
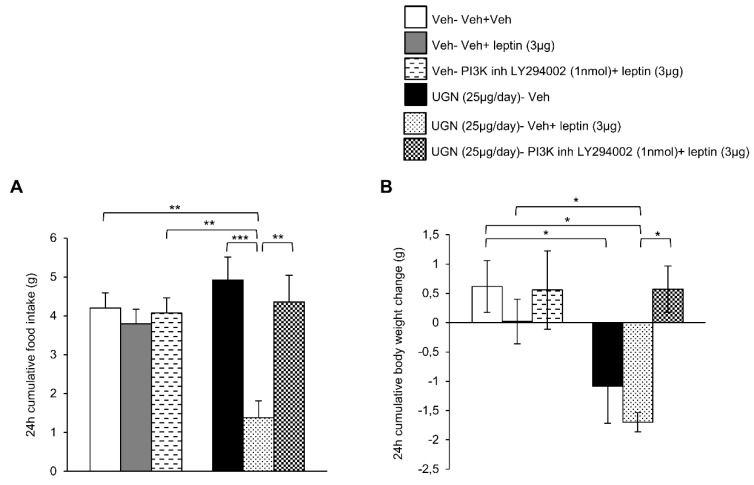
Evaluation of the blockade of PI3K pathway in the response in 24 h cumulative food intake (**A**) and body weight change (**B**). The mice were perfused with UGN (25 µg/mouse) for four days, submitted to fasting overnight and then injected with leptin (3 µg/mouse), an inhibitor of PI3K (LY294002) (1 nmol/mouse), or by the combination leptin/LY294002. Values are represented as mean ± SEM; *n* = 7–8 animals per group. * *p* < 0.05; ** *p* < 0.01, *** *p* < 0.001 vs. vehicle.

**Figure 5 nutrients-11-00752-f005:**
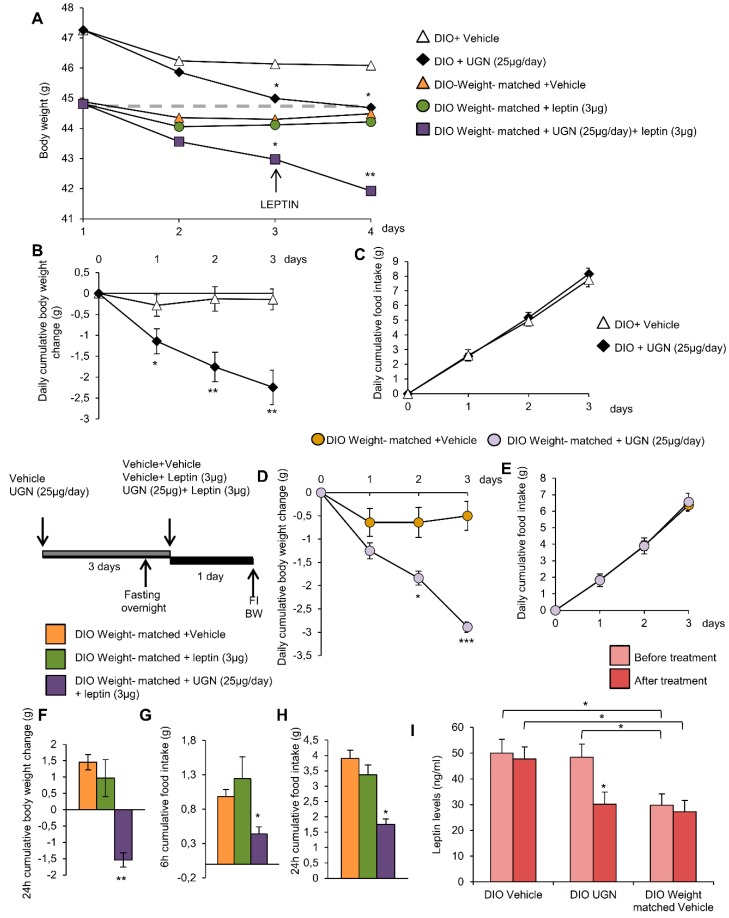
Effect of chronic ICV UGN (25 µg/mouse) injection for four days and ICV leptin (3 µg/mouse) for 24 h in DIO mice and weight-matched DIO mice on body weight (**A**). Effect of chronic ICV saline or UGN (25 µg/mouse) injection for three days on body weight change (**B**) and food intake (C) in DIO mice. Effect of chronic ICV saline or UGN (25 µg/mouse) injection for three days on body weight change (**D**) and food intake (**E**) in weight-matched DIO mice. Effect of ICV leptin (3 µg/mouse) infusion in weight-matched DIO mice (the mice being pretreated with UGN (25 µg/mouse/four days) and submitted to fasting the night before leptin injection) on a 24 h body weight change (**F**), six and 24 h cumulative food intake (**G** and **H** respectively). Circulating leptin levels in DIO mice and weight-matched DIO mice treated with ICV vehicle or UGN (**I**). Values are represented as mean ± SEM; *n* = 5–14 animals per group. * *p* < 0.05; ** *p* < 0.01, *** *p* < 0.001 vs. vehicle.

**Table 1 nutrients-11-00752-t001:** Primers and probes used for gene amplification.

Gene	Direction	Primer Sequence 5’ → 3’
HPRT	FWD	AGCCGACCGGTTCTGTCAT
	REV	GGTCATAACCTGGTTCATCATCAC
	PB	CGACCCTCAGTCCCAGCGTCGTGAT
AgRP	FWD	GCACAAGTGGCCAGGAACTC
	REV	CAGGACACAGCTCAGCAACAT
	PB	CAAGCATCAACAAGCAAAGGCCATGC
POMC	FWD	CGTCCTCAGAGAGCTGCCTTT
	REV	TGTAGCAGAATCTCGGCATCTTC
	PB	CGGGACAGAGCCTCAGCCACC

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
