# Peer review of "Uroguanylin Improves Leptin Responsiveness in Diet-Induced Obese Mice"

_nutrients, 2019, doi:10.3390/nu11040752_

Round 1

Reviewer 1 Report

This manuscript is significantly improved with the addition of data showing that the effect of UGN on leptin sensitivity is not due to decreased leptin levels as a result of decreased weight.  The results are consistent with earlier findings and important insight into mechanism are provided. 

The authors have added acknowledgement of the use of pharmacologic doses of UGN, which likely does not reflect endogenous levels that would reach the brain. Nevertheless, the findings do support that UGN can influence leptin sensitivity.  As a potential therapeutic approach, this provides rationale for further investigation. 

Author Response

Reviewer 1

REPLY: The manuscript has been entirely edited by an English native speaker and we hope that now it reads well.

Reviewer 2 Report

This paper could be of great interest to the field of obesity however, it is difficult to follow. Some of the figures (in particular, Figure 3) are not easily understandable, especially regarding the representation of the statistical significance (it is unclear where the bars start and stop). I would advise to review the presentations of the figures, especially the complex ones when colours are used.

It is also unclear why there is a difference in animals and where it applies - clarity should be provided.

The description/discussion should be refined especialy regarding the molecular mechanisms.

The English should be extensively edited: some sentences do not really make sense,

Author Response

Reviewer 2

Comments and Suggestions for Authors

This paper could be of great interest to the field of obesity however, it is difficult to follow. Some of the figures (in particular, Figure 3) are not easily understandable, especially regarding the representation of the statistical significance (it is unclear where the bars start and stop). I would advise to review the presentations of the figures, especially the complex ones when colours are used.

REPLY: We thank the Reviewer for her/his encouraging comments and recommendations. We have now changed the presentations of Figures 3, 4 and 5 and hopefully are now clear to the readers.

It is also unclear why there is a difference in animals and where it applies - clarity should be provided.

REPLY: Again, we hope that with the new presentations the figures are clear and there are no doubts about where differences apply.

The description/discussion should be refined especialy regarding the molecular mechanisms.

REPLY: We have rewritten the discussion and changed the part related to STAT3 and PI3K, which were the targets studied here. Our data indicate that PI3K is the responsible for the increased leptin responsiveness when DIO mice receive uroguanylin.

The English should be extensively edited: some sentences do not really make sense,

REPLY: The manuscript has been entirely edited by an English native speaker and we hope that now it reads well.

This manuscript is a resubmission of an earlier submission. The following is a list of the peer review reports and author responses from that submission.

Round 1

Reviewer 1 Report

This is a straightforward study that examines the effects of uroguanylin (UGN) and leptin on food intake and body weight  in a rodent model of high fat diet-induced obesity.  The authors specifically test the hypothesis that centrally administered UGN can restore leptin sensitivity.  Although the study design is generally appropriate and the data generated can support the conclusions made, it is not clear whether this is directly due to UGN or an indirect consequence of reduced leptin levels from UGN-induced weight loss.  In the experimental data, leptin levels before and after the 4 days of UGN treatment should be reported.  If there is indeed a difference, this needs to be addressed in the discussion.  Another relevant control would be to examine the effect of leptin administration in animals that are weight matched independent of UGN exposure.

In Figure 1 panel B, UGN is shown to cause a progressive decrease in body weight while this is administered.  However, in the UGN control in panel D, there is no subsequent weight decrease.  It is not clear why this is the case.  

It is understood that the goal of this study is to provide data that may potentially be useful in pharmacotherapy. Nevertheless, it would be helpful to at least mention the relative amounts of UGN used in these experiments to those that might be achieved from physiologic sources.